# Bibliometric Mapping of Academic Research Focusing on Animal Production and Climate Change in Association with Methane Emissions and Animal Productivity

Akeem Babatunde Sikiru [1], Olayinka John Makinde [2], Bossima Ivan Koura [3],*,
Stephen Sunday Egena Acheneje [4], John Olushola Alabi [4,5], Maria Ndakula Tautiko Shipandeni [6]
and Oludayo Michael Akinsola [7]

[1] Department of Animal Science, Federal University of Agriculture Zuru, Zuru 872101, Nigeria;
akeembaba01@gmail.com
[2] Department of Animal Science, Faculty of Agriculture, Federal University Gashua, Gashua 671105, Nigeria;
johyinmak@yahoo.com
[3] Ecole de Gestion et d'Exploitation des Systèmes d'Elevage, Université Nationale d'Agriculture,
Ketou P.O. Box 43, Benin
[4] Department of Animal Production, Federal University of Technology Minna, Minna 920101, Nigeria;
acheneje.egena@futminna.edu.ng (S.S.E.A.); alabioj@gmail.com (J.O.A.)
[5] Research Office, University of Limpopo, Private Bag X1106, Sovenga 0727, South Africa
[6] Department of Animal Production, Agribusiness and Economics, University of Namibia,
Windhoek 13301, Namibia; tautiko@gmail.com
[7] Department of Veterinary Medicine, Theriogenology and Production, University of Jos, Jos 930103, Nigeria;
dayoakinsola@gmail.com
* Correspondence: kouraivan@gmail.com

**Abstract:** Climate change is a pressing global challenge, and animal production is a major contributor to methane emissions. This study examines the academic landscape of research on $CH_4$ emissions and animal productivity, with a focus on cattle, sheep, and goats. Using a bibliometric analysis of 2500 documents published between 1987 and 2023, the study finds that research on this topic has increased significantly over time, with a record high in 2022. The leading countries in terms of research output are the United States, China, Brazil, Canada, and Italy. The study identifies several key research themes, including the impact of $CH_4$ emissions on animal productivity parameters, the development of mitigation strategies, and the assessment of trade-offs and synergies between $CH_4$ emissions reduction and other sustainability goals. The study concludes by highlighting the importance of continued research on $CH_4$ emissions and animal productivity to develop and implement effective mitigation strategies. This study has important implications for policymakers, researchers, and the livestock industry. Policymakers can use the findings to inform the development of policies and regulations that support the reduction of $CH_4$ emissions from animal production. Researchers can use the findings to identify gaps in the existing knowledge base and to develop new research directions. The livestock industry can use the findings to develop more sustainable production practices. By working together, policymakers, researchers, and the livestock industry can develop and implement effective mitigation strategies that reduce greenhouse gas emissions, protect the environment, and support sustainable food production.

**Keywords:** climate change; methane emissions; animal production; sustainability

## 1. Introduction

Climate change is one of the most pressing global challenges, with far-reaching consequences for ecosystems, economies, and well-being [1,2]. Meanwhile, greenhouse gas emissions, particularly methane ($CH_4$) associated with animal production contribute to this global crisis [3]. Hence, as the planet struggles with the urgent need to mitigate the

impact of climate change, academic research has a pivotal role in driving evidence-based policy decisions for shaping the discourse surrounding climate change as a critical issue. In acknowledgment of this situation, this study was carried out to delve into the complex and interconnected realms of animal production, climate change, and $CH_4$ emissions. The study objective was to examine the growth and trends of published documents associated with animal production and climate change with a specific emphasis on $CH_4$ emissions and the production of common food animals, including cattle, sheep, and goats, across different parts of the world. This is because the comprehensive exploration of the academic landscape in regard to this subject matter holds key roles in charting growth and trends of research on animal production and climate change [4]. The perspective could contribute to uncovering the dynamics that can help shape future research on animal production concerning climate change.

This is important because, in the quest for a sustainable future, it is imperative to assess the impact of academic research on policy formulation [5]. This justifies the need to scrutinize scholarly scientific performance and evaluate tangible effects on both animal production practices and climate change policies. Through rigorous analysis and a commitment to scientific inquiry, the paper aims to shed light on the past, present, and future of academic research's role in addressing this critical issue. This is to provide valuable guidance for decision-makers and researchers alike, fostering the development of sustainable policies and practices that can mitigate the impacts of animal production and climate change on our planet. The study identified trends in documents published over the years on livestock production and climate change, pinpointed research gaps in the existing literature, and provided guidance for future research in the areas of animal production and climate change. These will contribute to knowledge that can help to steer future research efforts towards more informed and impactful directions.

## 2. Materials and Methods

### 2.1. The Search Queries and Research Database

The study implemented a comprehensive bibliometric analysis of research related to the interplay between livestock production and climate change aimed to uncover trends, patterns, and the evolution of research using the Dimensions database. The database was selected as the primary source of data for the study because it is a multidisciplinary research database providing access to a vast array of scholarly publications, patents, and grant data. The search query used for this study includes "methane emissions and animal productivity" AND "sheep, cattle, and goat and climate change". "Methane emissions and animal productivity" referred to enteric methane emissions from ruminant animals. "Livestock production and climate change" referred to how livestock production is affected by or enhances climatic changes. These were used as keywords for the purpose of creating specific relationships between livestock production, methane emission, and animal productivity.

Animal Productivity refers to the efficiency with which an animal converts feed into products like meat, milk, eggs, or wool in terms of the following:

- Growth rate: How quickly an animal gains weight or produces milk;
- Feed conversion ratio: The amount of feed needed to produce a unit of product (e.g., grams of feed per kilogram of meat);
- Reproductive performance: The number of offspring an animal produces.

Methane is a greenhouse gas 25 times more efficient at warming the planet than carbon dioxide over a 100-year period. In animals, it is mainly produced by ruminants (cattle, sheep, and goats) during digestion in their stomachs. Microbes ferment plant material, releasing methane as a byproduct, which was why this study focused on cattle, sheep, and goats.

Animal productivity and methane emissions have a complex relationship. While higher productivity often requires more feed intake, leading to potentially higher methane emissions, strategies like improved feed quality and digestibility can increase productivity while reducing methane production per unit of product.

Livestock production (as Ruminant Production) refers to the raising and breeding of any animals for food or other products, but in this context, it specifically refers to raising ruminant animals, such as cattle, sheep, and goats. It encompasses all aspects of their care, from breeding and feeding to housing and transportation.

Climate Change is related to the long-term alteration of temperature and typical weather patterns in a place. Climate change could be understood by changes in the following:

- Temperature: Rising average temperatures and more extreme heat events.
- Precipitation: Changes in rainfall patterns, including droughts and floods.
- Sea level: Rising sea levels due to melting glaciers and ice sheets.

Livestock production can be linked to climate change in several ways:

- Methane emissions: As mentioned earlier, ruminants release methane during digestion.
- Land use: Raising livestock requires land for grazing and feed production, which can lead to deforestation and habitat loss.
- Manure management: Improper manure management releases methane and nitrous oxide, another potent greenhouse gas.

Therefore, publications and other scholarly works investigating the relationship between $CH_4$ emissions from livestock, their impact on animal productivity parameters (growth rate, reproduction, overall performance), and sheep, cattle, and goat production in the context of a changing climate were captured. To ensure the relevance and accuracy of the publication data, full-text scholarly articles, original research articles, and reviews in peer-reviewed journals published in English were used as inclusion criteria.

### 2.2. Data Retrieval

A comprehensive search of the Dimension database app.dimensions.ai (accessed on 20 October 2023) was conducted to explore the relationship between "$CH_4$ emissions and animal productivity" and "sheep, cattle, and goat production and climate change". To retrieve pertinent articles with these queries, the following search queries were devised:

1. For "$CH_4$ emissions and animal productivity", there was a search for articles that were published from 1980 to 2023.
2. For "sheep, cattle, and goat production and climate change", there was a search for articles that were published from 1980 to 2023.

The search queries were designed to encompass a wide range of the literature in the field of animal production and its association with $CH_4$ emissions. After the initial data retrieval, a data cleaning process was implemented to ensure the quality and relevance of the retrieved articles. This includes activities such as the removal of duplicates and irrelevant records and any articles falling outside the specified publication date range while the final dataset was used for subsequent analysis, making use of articles meeting these criteria. This data retrieval and cleaning process aimed to create a high-quality dataset that would serve as the foundation for the comprehensive bibliometric mapping analysis.

### 2.3. Bibliometric Analysis Mapping and Visualization

The bibliometric analysis was carried out using VOSviewer which is a widely recognized and user-friendly software designed for visualizing and analyzing bibliometric data. It is particularly well-suited for creating visual representations of scholarly networks, identifying research trends, and exploring relationships between research papers, authors, keywords, and journals. Prior to using VOSviewer, the data collected from the Dimensions database were organized and cleaned to ensure their compatibility with the software. This involved the extraction of key metadata from each publication, including title, authors, publication year, source, keywords, and citation counts. VOSviewer allows for the easy import of bibliometric data. The datasets were prepared in CSV and Excel prior to their importation into VOSviewer, which handled both the bibliographic and citation data seamlessly.

The VOSviewer was used to construct bibliometric networks based on relationships between various entities within the datasets. In the analyses, co-occurrence networks were

generated to visualize the associations between research papers, authors, and keywords. Specifically, the VOSviewer was used to generate networks for each phrase where the strength of the connection between papers was determined by the number of common references they shared. The various analytical functions provided by VOSviewer were used to gain more insights into the bibliometric data, including cluster analysis to identify groups of closely related research papers. The data obtained from the VOSviewer analysis were used to identify country-wise contributions to documents published as well as the total citations to draw meaningful conclusions and insights regarding the trends, key research themes, and collaboration patterns on the search phrases. The VOSviewer was used to harness the software's capabilities for bibliometric data visualization and analysis, enabling a comprehensive exploration of the research landscape related to $CH_4$ emissions and animal production.

## 3. Results and Discussion

### 3.1. Number of Documents Published and Total Citations Associated with Methane Emissions and Animal Productivity between 1987 and 2023

Between 1987 and 2023, the average number of published documents on methane ($CH_4$) emissions and animal productivity was 67.56 per year; the minimum and maximum published documents were 0 and 675, respectively. The total number of published documents for the 37-year period was 2500. This showed increasing trends in the number of documents published annually; the total citations starting from 2002, compared with the years 1987–2000, continues to increase with record-high total citations and documents published falling between 2009 and 2022, respectively (Figure 1).

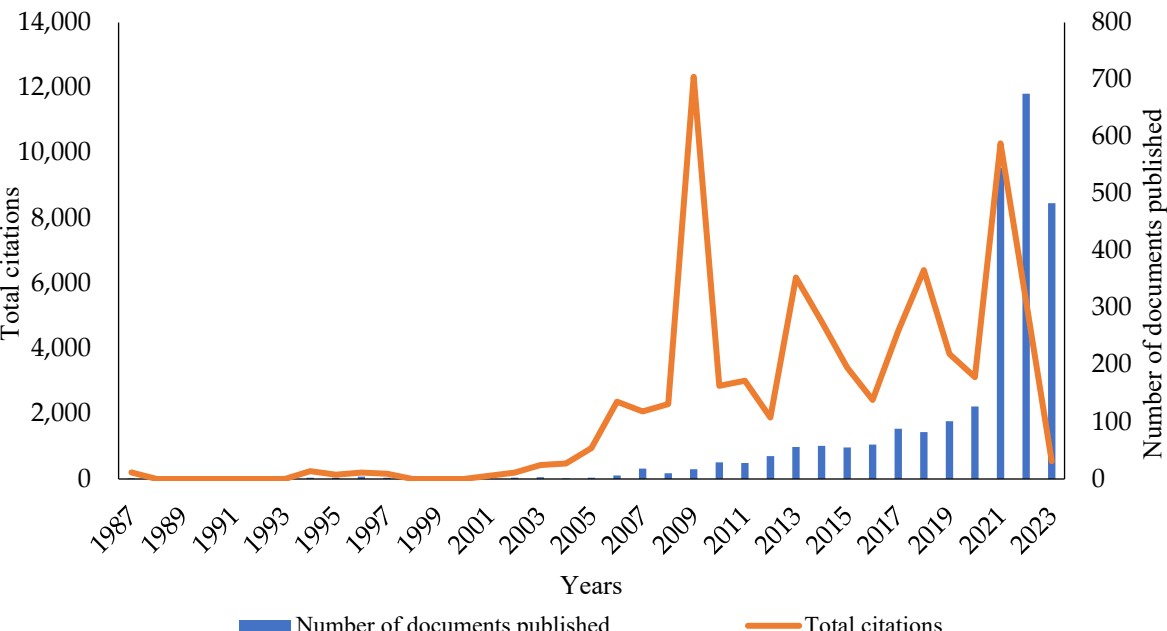

**Figure 1.** Trend in number of documents published in relation to total citation on $CH_4$ emissions and animal productivity (1987–2023). The figure illustrates the temporal evolution of research on $CH_4$ emissions and animal productivity, including growth rate, feed conversion ratio (FCR), milk yield, reproduction rate, dressing percentage, mortality rate, fertility rate, age at puberty, health and disease resistance, product quality, livestock reproductive efficiency, and animal behavior from 1987 to 2023. Some of the publications highlighted exploration of feeding and production management as means of abating methane emissions in livestock production. The *x*-axis represents the chronological time frame, while the *y*-axis shows the total number of documents published and total citations. The blue line graph demonstrates growth in the number of documents published in the specified field over

this period. The overlaid orange lines represent the cumulative total of citations received by these publications. The figure reveals the relationship between the growth of research output and its corresponding impact, as measured by the accumulation of citations, providing insights into the research landscape's development and scholarly influence over time.

The bibliometric analysis revealed a notable evolution in research on the relationship between climate change and livestock production. Before 2002, there was a relatively low average number of published documents on $CH_4$ emissions and animal productivity, indicating limited attention being paid to research in this area. However, from 2002 onwards, there has been a clear and consistent increase in the number of publications on the subject matter, reflecting a growing recognition of the importance of understanding the interplay between climate change and livestock production. This rise in research activity aligns with a broader global awareness of climate change and its implications on livestock production [6]. The bibliometric analysis covering the period from 1987 to 2023 focused on the relationships between $CH_4$ emissions and animal productivity; it provides a valuable glimpse into the evolution of research in this field and its broader implications. The data indicate a notable growth in research activity over the years. The relatively low average number of published documents (67.56 per year) before the year 2002 suggests that research in this area was not receiving significant attention. However, from 2002 onwards, there was a clear and consistent increase in the number of publications, pointing to a growing recognition in the field. This rise in research activity aligns with a broader global awareness of climate change and the environmental impact of agriculture, specifically livestock farming.

This study underscores the importance of understanding the relationship between $CH_4$ emissions and animal productivity, as it is evident from the records of published documents. Some of these publications suggested strategies such as feeding management, the addition of inhibitors, and vaccination as techniques to change the population of rumen methanogens, which are the main producers of $CH_4$ in animals [7,8]. Additionally, the composition of the microbiota in the rumen was found to have a major influence on $CH_4$ emissions in some of the publications [7,9]. The contrasting effects of different diets, such as a high-concentrate diet and low-quality grass forage, on the composition of rumen microbiota and $CH_4$ yield were highlighted as the need for tailored mitigation strategies. The upward trend in research activity also reflects the growing concern over $CH_4$ emissions since it is a potent greenhouse gas with substantial emissions coming from livestock, particularly ruminant animals.

An understanding and the mitigation of these emissions have become a critical aspect of addressing climate change. This was attested to by the increasing research conducted during this period, which played significant roles in the development of strategies and technologies to reduce $CH_4$ emissions from livestock while maintaining and enhancing animal productivity. The range of citations per year is striking, with some studies attracting thousands of citations while others garnered little attention. This highlights the diversity of research efforts and the varying degrees of influence individual studies have had on the field of $CH_4$ emissions and animal productivity. The highly cited papers likely introduced innovative approaches or made groundbreaking discoveries that could shape the direction of future research and informed policy decisions [10]. The years 2009 and 2022, with their record numbers of citations and publications, respectively, are of particular interest in the research on $CH_4$ emission and animal productivity. In the year 2009, at the end of the first decade of the 21st century, there could be studies that may have presented novel findings on $CH_4$ emissions and animal productivity. The introduction of novel technologies for measuring $CH_4$ emissions in animals in 2009 could have been a significant step forward, leading to an increase in published documents for the year. This could be because of the introduction of novel technologies for measuring $CH_4$ emissions in animals, such as laser-based detectors, $CH_4$ sniffing dogs, $CH_4$-sensitive cameras, green feed systems, and SF6 gas tracer techniques. This offered more accuracy and sensitivity in measuring $CH_4$ concentrations from individual animals or large groups. This has the potential to improve

the understanding of how $CH_4$ is emitted from animals and to develop strategies to reduce $CH_4$ emissions.

### 3.2. Global Mapping and Clustering of $CH_4$ Emissions and Animal Productivity Research Publications

The visualization analysis of the datasets on emissions and animal productivity obtained from Dimensions app.dimensions.ai (accessed on 20 October 2023) showed that the documents published on the subject matter were from 104 countries across the world. These countries are in eight clusters based on the strength and number of publications per country (Figure 2). The clusters are led by countries such as China, Brazil, Canada, Italy, South Africa, the United States, France, and Australia, and the average publications per member of the cluster are 39.74, 20.82, 64.00, 26.92, 12.55, 134.43, 34.17, and 49.00 documents, respectively (Table 1).

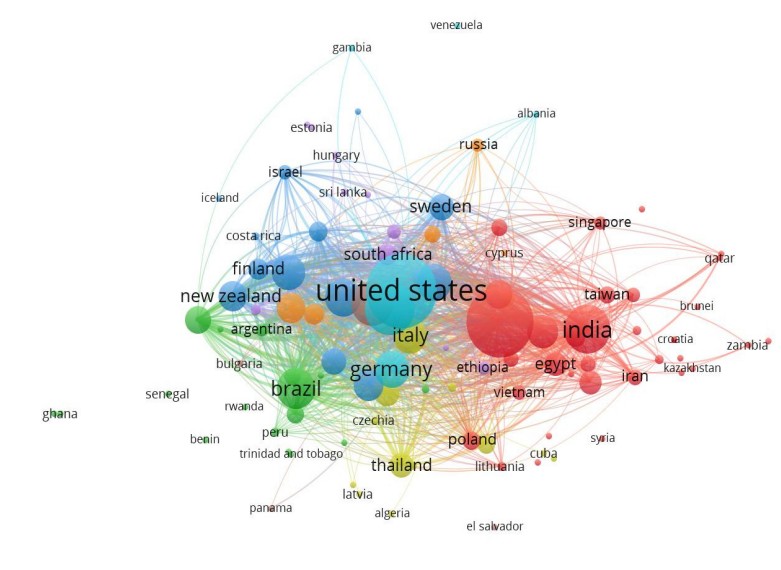

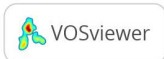

**Figure 2.** Clustering of countries publishing documents on $CH_4$ emissions and animal productivity. This figure showcases the clustering of countries engaged in producing documents related to $CH_4$ emissions and animal productivity. The clustering is based on the origin and affiliations of authors, spanning from 1987 to 2023. Notably, eight distinct clusters emerged, each spearheaded by a prominent country: China, Brazil, Canada, Italy, South Africa, the United States, France, and Australia. These clusters visually represent the global distribution of research contributions, highlighting the leading nations in this field during the specified time frame.

**Table 1.** Clustered research activity on $CH_4$ emissions and animal productivity in relation to average published documents across the world (1980–2023).

| Clusters | Average Published Document per Member | Clusters Lead |
|----------|---------------------------------------|---------------|
| 1        | 39.74                                 | China         |
| 2        | 20.82                                 | Brazil        |
| 3        | 64.00                                 | Canada        |
| 4        | 26.92                                 | Italy         |

**Table 1.** *Cont.*

| Clusters | Average Published Document per Member | Clusters Lead |
|---|---|---|
| 5 | 12.55 | South Africa |
| 6 | 134.43 | United States |
| 7 | 34.17 | France |
| 8 | 49.00 | Australia |

*3.3. Country-Wise Document Publication and Average Citations per Document Published on Methane Emissions and Animal Productivity between the Years 1987 and 2023*

The United States topped the list of countries from which the documents are being published during the years under study, with more than 500 published documents. However, the citation per document from the United States published was lower compared to some countries where a lower number of documents were published during the period (Figure 3).

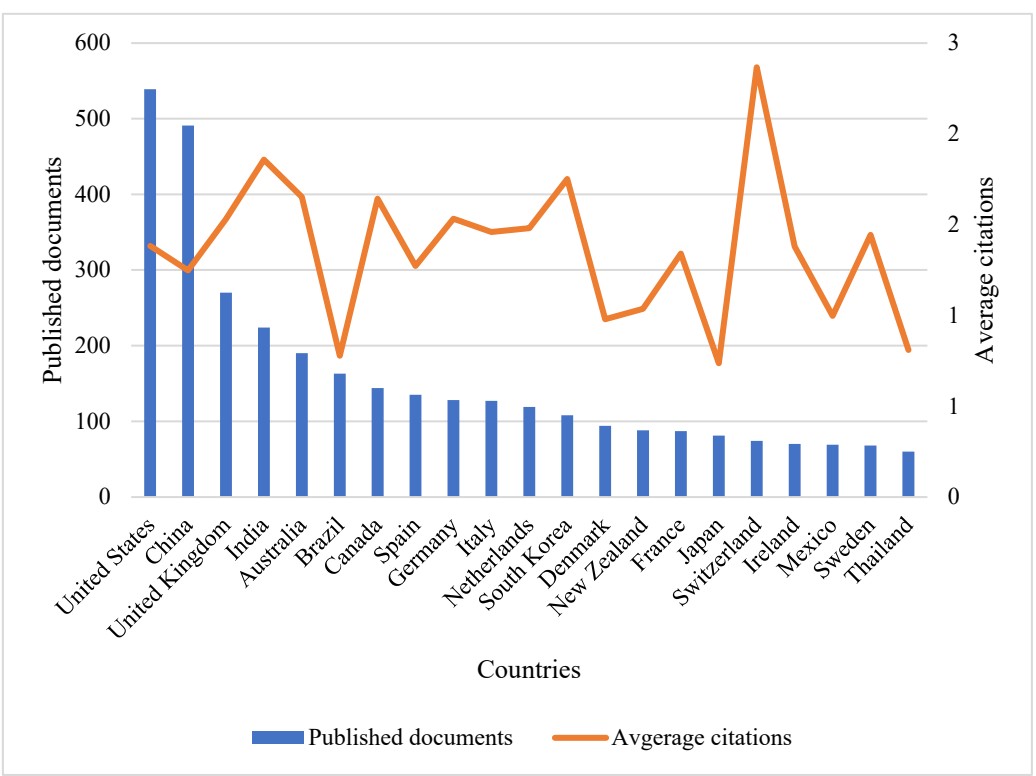

**Figure 3.** Leading countries for publications on $CH_4$ emissions and animal productivity (1987–2023). This figure illustrates the leading countries that contributed to research on the interrelation between $CH_4$ emissions and animal productivity over the years 1987 to 2023. The height of the bar graph is proportionate to the number of documents published in the field and the average annual citation per document published from each of the countries. The chart provides valuable insights into the global distribution of research efforts related to this topic and identifies the key countries that significantly contributed to $CH_4$ emissions and animal productivity.

The specific leadership of the United States in the subject matter could be linked to activities such as research findings at some US universities that made use of laser-based $CH_4$ detectors to measure $CH_4$ emissions from individual dairy cows [8]. These researchers found that $CH_4$ emissions varied significantly between individual cows and that the highest emissions came from cows with the highest milk production [9]. Moreover, in the United States, researchers at the University of New Hampshire have been among the centers of

excellence championing research on $CH_4$ mitigation using dietary options in cattle and maintaining a strong contribution to the field up till the golden era of published documents on $CH_4$ emission, which was the year 2009, and even after the period [10]. In 2022, the record number of publications suggests a surge in research activity, possibly spurred by increased global awareness of climate change, a growing emphasis on sustainable agriculture practices, and the need to focus on possible climate change-induced challenges in the post-COVID-19 era. The high number of citations and the overall influence of the research on $CH_4$ emission and animal productivity indicate its potential to inform policy decisions and influence agricultural practices.

Therefore, policymakers seeking to address climate change may turn to this body of research for evidence-based solutions for making decisions on research directions and policy formulations, while farmers may implement the recommendations to reduce $CH_4$ emissions, thereby improving their livestock's productivity. The data obtained through the bibliometric analysis on $CH_4$ emission and animal productivity strongly support the ongoing relevance of research associated with $CH_4$ emissions and animal productivity. With the status of climate change as a global challenge and the need for sustainable agricultural practices growing, the importance of this research is likely to persist. Furthermore, the upward trend of the research suggests that the field will continue to expand, with an increasing number of researchers and organizations recognizing its significance and the need to get involved. This study identified and suggested that $CH_4$ emissions and animal productivity are both increasingly prominent and influential fields of research because they could have implications extending across climate change mitigation, sustainable agriculture, and policy development. As the world continues to grapple with environmental challenges, the insights gained from this research will likely remain vital in shaping our responses and strategies for the future.

### 3.4. Document Trends in Livestock Production and Climate Change

The total number of published documents for the 43-year period (1980–2023) focusing on sheep, cattle, and goat production concerning climate change was 4472. The results showed an increasing trend in the number of documents published and total citations starting from 2005. Although there was a slow rise in the number of documents published and total citations per year until 2008, when the total citations increased, the number of documents published remained low until the year 2021, when the number of published documents increased from 92 (in the year 2020), to record high number of 757 published documents (Figure 4).

The trend in published documents on sheep, cattle, and goat production in relation to climate change from 1980 to 2023 carries significant implications. The rising number of publications and total citations, particularly since 2005, underscores the growing recognition of the critical link between livestock production and climate change. This suggests a concerted effort by the research community to better understand and address the environmental impact of livestock, highlighting its importance as a research area. This is in line with the general agreement among scientific reports that livestock production has a significant impact on climate change because, according to the Intergovernmental Panel on Climate Change (IPCC), livestock production is responsible for 14.5% of global greenhouse gas emissions [11]. The abrupt surge in the number of documents published in 2021, with 757 publications compared to 92 the previous year, could signify a potentially transformative moment in the field of research. This surge could likely serve as an indicative response to emerging challenges, the adoption of innovative research approaches, or increased awareness of the pressing need to mitigate the environmental consequences of livestock production post-COVID-19. This type of growth in research output has the potential to inform policies, guide sustainable agricultural practices, and raise awareness among the broader public regarding the role of livestock in climate change, further emphasizing the field's evolving nature and its real-world implications.

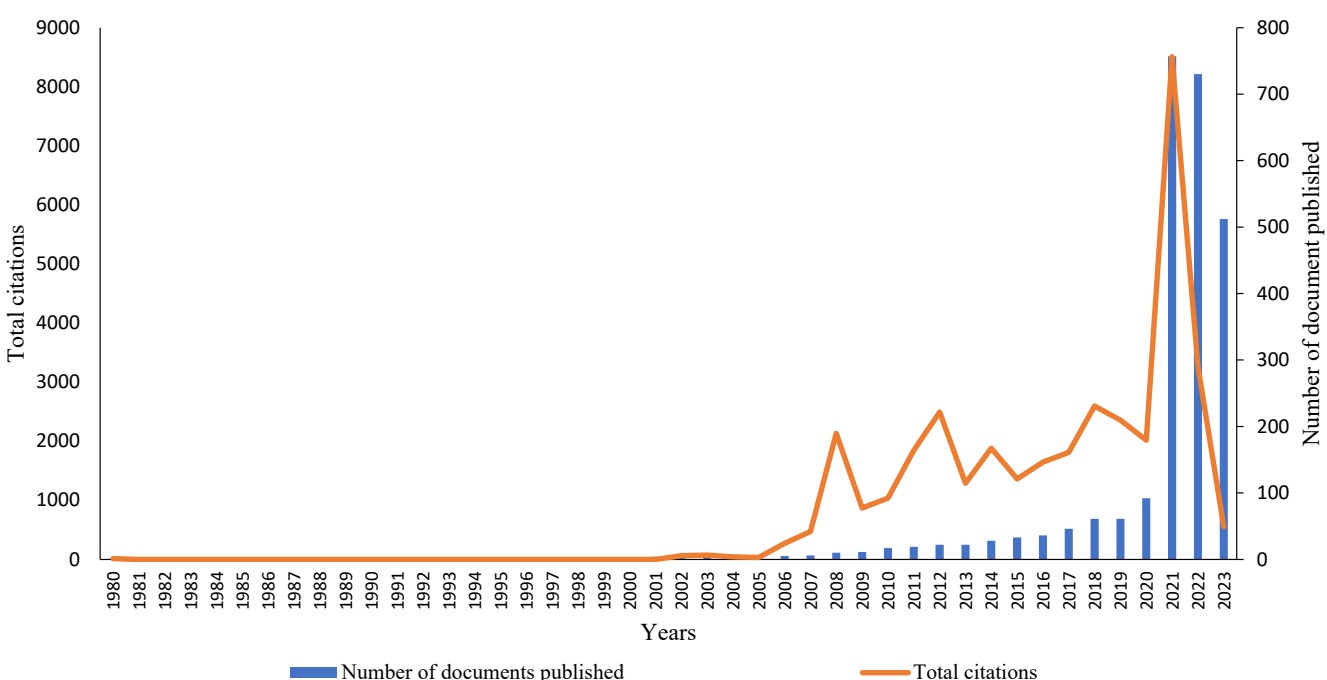

**Figure 4.** Trend in number of documents published and citations on sheep, cattle, and goat production (1980–2023). The figure illustrates the temporal evolution of scholarly activity and research in the field of livestock production, focusing on sheep, cattle, and goat species. The graph spanned the years from 1980 to 2023, offering a comprehensive overview of the publication and citation trends over this period. The blue line represents the number of documents published, reflecting the annual quantity of research contributions pertaining to these livestock categories. The trendline revealed fluctuations and growth patterns in research output depicting the cumulative number of citations received by these documents; the citations are a key indicator of the impact and influence of the research. The trendline's trajectory provides insights into the research's reception and influence within the scientific community. The figure also highlights notable shifts (rising or declining trends) and major milestones in the accumulation of knowledge and scholarly engagement related to sheep, cattle, and goat production in the context of changing climate during the specified time frame as valuable information for understanding the evolving landscape of research in this field.

### 3.5. Global Publication Trends on Livestock and Climate Change: A 138-Country Analysis

The documents published on sheep, cattle, and goats and climate change were published from 138 countries clustered into 10, each cluster led by countries including the United States, India, Brazil, Belgium, China, Spain, Canada, Australia, France, and Italy (Table 2). The total number of documents published on the subject matter over the duration of 43 years was 4472, while the average document published per each participating country was 32.40. The minimum and maximum published per country were 1 and 411 documents, respectively. Prominent among the nations are the United States, Italy, Spain, China, India, Brazil, and others (Figure 5).

**Table 2.** Clustered research activities on sheep, cattle, and goats and climate change in relation to average published documents across the world (1980–2023).

| Clusters | Average Published Document per Member | Clusters Lead |
|---|---|---|
| 1 | 39.24 | United States |
| 2 | 32.82 | India |
| 3 | 23.65 | Brazil |
| 4 | 19.05 | Belgium |
| 5 | 36.06 | China |

**Table 2.** *Cont.*

| Clusters | Average Published Document per Member | Clusters Lead |
|---|---|---|
| 6 | 31.09 | Spain |
| 7 | 18.00 | Canada |
| 8 | 48.60 | Australia |
| 9 | 31.75 | France |
| 10 | 104.33 | Italy |

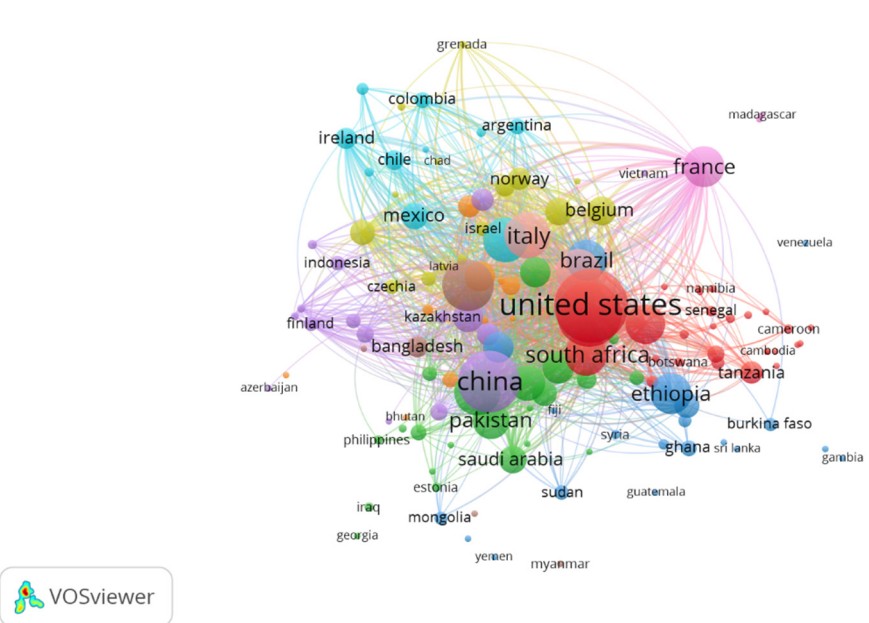

**Figure 5.** Clustering of countries publishing documents on sheep, cattle, and goat production and climate change are being published from 1980 to 2023. This figure showcases the clustering of countries engaged in producing documents related to animal production and climate change. The clustering is based on the origin and affiliations of authors, spanning from 1980 to 2023. Notably, 10 distinct clusters emerged, each being led by prominent countries, including the United States, India, Brazil, Belgium, China, Spain, Canada, Australia, France, and Italy. These clusters visually represent the global distribution of research contributions, highlighting the leading nations in this field between 1980 and 2023. There was a diverse range of research from these countries, cutting across sheep, cattle, and goats, investigating the impact of climate change on land use and forage and feed scarcity. Some of the studies explore how changing precipitation patterns lead to droughts, floods, and land degradation, necessitating sustainable land management practices like rotational grazing. Some research also examines feed scarcity due to rising temperatures and altered precipitation, prompting exploration of heat-resistant forage varieties, alternative feed sources, and precision agriculture. These studies consider both global and regional contexts, highlighting the need for mitigation strategies in developed countries and adaptation strategies for resource-poor farmers in developing ones. Overall, the research output during the period 1980–2023 focuses on the sustainability of livestock production under changing climate conditions.

The publication of documents related to the impact of sheep, cattle, and goats on climate change from 138 different countries underscores the global significance of climate change and livestock production. It also reflects concern and awareness associated with the general societal and environmental implications of livestock production. The fact that research on this topic is distributed across such a large number of countries implies a collective recognition of the need to address the environmental challenges associated with livestock farming on a global scale. The clustering of these countries into groups, each

led by prominent nations like the United States, India, and Brazil, suggests that there are established collaborative networks and regional research interests on the subject matter. This indicates that researchers from various countries are working together to understand the interplay between livestock and climate change, sharing knowledge and resources, and potentially driving global initiatives for sustainable livestock practices. However, compared to others, the disparities in research output highlight the need for efforts to promote equitable participation and knowledge exchange on climate change and livestock production. This agrees with the report of a scientific outcome of an IPBES-IPCC co-sponsored workshop on biodiversity and climate change convened in Switzerland [12]. Overall, this underscores the urgent need for international cooperation to mitigate the environmental impacts of livestock production while also addressing disparities in research capacity.

*3.6. Global Leadership in Livestock–Climate Change Research: A Citation Perspective*

The United States tops the list of countries where documents on the production of sheep, cattle, and goats concerning climate change are being published. This was followed by the United Kingdom, China, Australia, and others. The United States also tops the list of countries with total citations (9078), followed by the United Kingdom (6201) and Kenya (4265) despite them publishing lower numbers of documents compared to other leading countries (Figure 6).

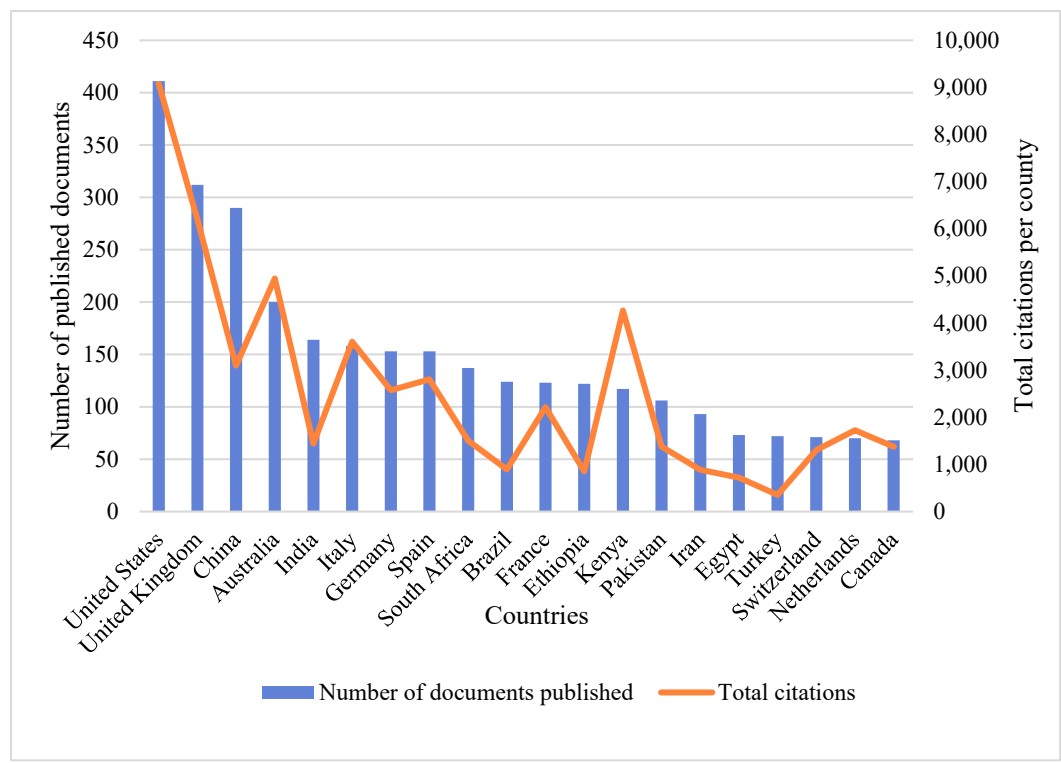

**Figure 6.** Countries where documents on sheep, cattle, and goat production in relation to climate change are being published from 1980 to 2023. This figure illustrates the leading countries that have contributed to research on the interrelation between livestock production and climate change. The height of the bar graph is proportionate to the number of documents published in the field and the total citations for all the documents published from each country. The chart provides valuable insights into the global distribution of research efforts covering animal production and knowledge sharing across the world.

## 4. Conclusions

The bibliometric analysis carried out in this study provides significant insights into the evolving landscape of research related to methane ($CH_4$) emissions, animal productivity,

and livestock production within the context of climate change. It reveals a noteworthy transformation in research activity in the fields over the years, with a clear shift from limited attention before 2002 to a consistent increase in publications and citations, reflecting the growing global recognition of the importance of this field of studies. This study is suitable to inform policy decisions and offers practical guidance for farmers in the quest for sustainable agricultural practices in a changing climate.

Likewise, the analysis of documents related to livestock production and its impact on climate change from 1980 to 2023 reveals a growing acknowledgment of the critical link between these factors. The rising number of publications and total citations, especially in recent years, signifies that there are concerted efforts by the global research community to better understand and address the environmental consequences of livestock production in relation to climate change. The sudden surge in publications in the year 2021 further reflects the dynamic nature of this research area, possibly prompted by emerging challenges, innovative approaches, or increased awareness post-COVID-19 pandemic. This growth has the potential not only to inform policies and sustainable practices but also to raise public awareness about the role of livestock in climate change, as well as the effect of climate change on livestock productivity. This study identifies and recommends the need for collaborative networks and regional research interests as a matter of urgency for promoting international cooperation to mitigate the environmental impacts on livestock production. Furthermore, as the world grapples with environmental challenges and the need for sustainable practices, the study suggests that research on livestock production and climate change remains pivotal in shaping responses and strategies for the future, touching upon climate change mitigation, sustainable agriculture, and policy development.

**Author Contributions:** Conceptualization, all authors; formal analysis, all authors; investigation, all authors; methodology, all authors; software, A.B.S. and O.J.M.; supervision, B.I.K. and O.M.A.; validation, all authors; writing—original draft, A.B.S.; writing—review and editing, all authors. All authors have read and agreed to the published version of the manuscript.

**Funding:** This research received no external funding.

**Data Availability Statement:** The datasets used and analyzed during the current study are available from the corresponding author upon reasonable request.

**Conflicts of Interest:** The authors declare no conflicts of interest.

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
