# Peer review of "Bibliometric Mapping of Academic Research Focusing on Animal Production and Climate Change in Association with Methane Emissions and Animal Productivity"

_ruminants, doi:10.3390/ruminants4010010_

Round 1

Reviewer 1 Report

Comments and Suggestions for Authors

Researchers and policymakers in the relevant field can gain valuable insights from bibliometric studies. Consequently, this study contributes to our understanding of methane emission and the relationship between animal productivity and climate change. An adequate amount of information and a clear statement of the study's purpose are provided in the introduction. There is clarity and coherence in the presentation of materials and methods. Results are communicated effectively. Certain statements must be referenced in the Discussion section. Upon incorporating the specified changes, the manuscript may be considered for publication.

Best regards

Major comments:

As "methane emissions and animal productivity" and "livestock production and climate change" starting years differ, it is crucial to explicitly distinguish between them in the materials and methods section.

Visual representations of the selection process in accordance with PRISMA guidelines enhance the study's comprehensibility.

Minor comments:

It is advisable to capitalize the initial letters of country names in the figures.

It is advisable to use "Average citation" instead of "Avg. citations" in tables and images.

L280: “… in some of the publications.” Kindly cite several sources.

L282: Kindly cite several sources.

L301-304: Please indicate the source for each example.

Author Response

Responses to Reviewer 1

Researchers and policymakers in the relevant field can gain valuable insights from bibliometric studies. Consequently, this study contributes to our understanding of methane emissions and the relationship between animal productivity and climate change. An adequate amount of information and a clear statement of the study's purpose are provided in the introduction. There is clarity and coherence in the presentation of materials and methods. Results are communicated effectively. Certain statements must be referenced in the Discussion section. Upon incorporating the specified changes, the manuscript may be considered for publication.

Best regards

Responses. Dear Reviewer, thank you very much for your nice words and interest in our study. We appreciate it.

Major comments:

As "methane emissions and animal productivity" and "livestock production and climate change" starting years differ, it is crucial to explicitly distinguish between them in the materials and methods section.

Responses. Explanation was provided, lines 82-90. The search was made for the same period and the approach was the same to evaluate both links, that is why we didn’t separate them. However, in the results a clear distinction was made.

Visual representations of the selection process in accordance with PRISMA guidelines enhance the study's comprehensibility.

Responses. Thank you dear editor.

Minor comments:

It is advisable to capitalize the initial letters of country names in the figures.

It is not possible to capitalize the initial letters of the country names because the figures were automatically generated from the software VOSviewer which was used to analyse the data which cannot be modified.

It is advisable to use "Average citation" instead of "Avg. citations" in tables and images.

Response. Done

L280: “… in some of the publications.” Kindly cite several sources.

Response: Done

L282: Kindly cite several sources.

Response: Done

L301-304: Please indicate the source for each example.

Response: Done

Reviewer 2 Report

Comments and Suggestions for Authors

Overall, the paper is well written and clear. Here you will find some minor and general comments. More specific comments are found in the attached file. 

The materials and methods, with all the software and methodologies used are explained clearly. 

The results are well explained with an important graphical work. 

The discussion is well written even if some of the topic touched, such as the different feeding strategies used to reduce methane weren't included in the results and analyses section. 

Also, the main results describes only the numerical changes in the publications and thus research activities done globally over time and in each country. If  this is surely interesting from a bibliographical point of view, however it is already stated and known since climate change and the role of livestock on it is well-researched from a long time ago. 

However, the paper is interesting, the process is very complex and might become more interesting if some aspects might be included. 

Here you have some main critical points that I have highlighted and some possible suggestions. 

1) a better explanation of which topics are included in the concepts of both "livestock production", "climate change" and also in terms of methane emissions. Climate change might include land use change, water-related issues, ghg emissions and so on. Production might refere to live weigh gain, milk production, health, digestive efficiency. I suggest you to state more correlty which topics did you include in the evaluation. Maybe you can include a table in the materials and methods. 

2) instead of only focusing on the number of papers related to those general topics, I suggest the Authors to include something related to the type of topics treated (nutritional strategies, production aspects....) in each country to to evaluate potential differences between different countries and also over time. it will help having a cleared pictures of the needs and goals of different countries. 

3) same when it comes to livestock production. Firstly it should be functional also to cluster the results and discussion by livestock species. Also, why including livestock species other than ruminants? 

You will find more specific comments in the pdf. 

Author Response

Responses Reviewer 2

Overall, the paper is well written and clear. Here you will find some minor and general comments. More specific comments are found in the attached file. 

The materials and methods, with all the software and methodologies used are explained clearly. 

The results are well explained with an important graphical work. 

Response. We thank the reviewer for his valuable comments and appreciation.

The discussion is well written even if some of the topic touched, such as the different feeding strategies used to reduce methane weren't included in the results and analyses section.

Response. The manuscript discussed studies conducted on methane emission and its links with animal productivity. However, we have added to the result section a sentence about various strategies to reduce methane that have been discussed in the used publications. Lines 182-183.

Also, the main results describes only the numerical changes in the publications and thus research activities done globally over time and in each country. If this is surely interesting from a bibliographical point of view, however it is already stated and known since climate change and the role of livestock on it is well-researched from a long time ago.

However, the paper is interesting, the process is very complex and might become more interesting if some aspects might be included. 

Response. Thank you, dear reviewer, for your comment. Yes, climate and the role of livestock is well-researched. Our manuscript explored the relationship between "methane emissions and animal productivity" and “Livestock production as affected by climate change (land use changes, manure management, etc.)” to identify current trends in research.

Here you have some main critical points that I have highlighted and some possible suggestions. 

  • a better explanation of which topics are included in the concepts of both "livestock production", "climate change" and also in terms of methane emissions. Climate change might include land use change, water-related issues, ghg emissions and so on. Production might refere to live weigh gain, milk production, health, digestive efficiency. I suggest you to state more correlty which topics did you include in the evaluation. Maybe you can include a table in the materials and methods. 

Responses. Details were provided. Please see line 80 – 115.

  1. Animal Productivity:

This refers to the efficiency with which an animal converts feed into products like meat, milk, eggs, or wool in terms of:

  • Growth rate: How quickly an animal gains weight or produces milk.
  • Feed conversion ratio: The amount of feed needed to produce a unit of product (e.g., grams of feed per kilogram of meat).
  • Reproductive performance: The number of offspring an animal produces.
  1. Methane Emissions:

Methane is a greenhouse gas 25 times more potent at warming the planet than carbon dioxide over a 100-year period. In animals, it's mainly produced by ruminants (cattle, sheep, and goats) during digestion in their stomachs. Microbes ferment plant material, releasing methane as a byproduct that was why this study focused on cattle, sheep, and goats.

  1. Animal Productivity and Methane Emissions:

These can have a complex relationship. While higher productivity often requires more feed intake, leading to potentially higher methane emissions, strategies like improved feed quality and digestibility can increase productivity while reducing methane production per unit of product.

  1. Livestock Production (as Ruminant Production):

This term can refer to the raising and breeding animals for food or other products. Still, in this context, it refers explicitly to raising ruminant animals like cattle, sheep, goats, and buffalo. It encompasses all aspects of their care, from breeding and feeding to housing and transportation.

  1. Climate Change:

This refers to the long-term alteration of temperature and typical weather patterns in a place. Climate change could cause changes in:

  • Temperature: Rising average temperatures and more extreme heat events.
  • Precipitation: Changes in rainfall patterns, including droughts and floods.
  • Sea level: Rising sea levels due to melting glaciers and ice sheets.
  1. Livestock Production and Climate Change:

Livestock production contributes to climate change in several ways:

  • Methane emissions: As mentioned earlier, ruminants release methane during digestion.
  • Land use: Raising livestock requires land for grazing and feed production, which can lead to deforestation and habitat loss.
  • Manure management: Improper manure management releases methane and nitrous oxide, another potent greenhouse gas.

instead of only focusing on the number of papers related to those general topics, I suggest the Authors include something related to the type of topics treated (nutritional strategies, production aspects....) in each country to evaluate potential differences between different countries and also over time. it will help having a cleared pictures of the needs and goals of different countries.

Response. Details about the types of topics treated were added in the results and discussion section, particularly, lines 182-183; lines 400 – 411.

same when it comes to livestock production. Firstly, it should be functional also to cluster the results and discussion by livestock species. Also, why including livestock species other than ruminants? 

Response. Thank you dear editor for your comment. Most of the studies used focused on ruminants. That is why we didn’t cluster by livestock species. There was so little information on the other species in the selected papers that we didn’t consider them. This was revised through the manuscript.

You will find more specific comments in the pdf. 

Page 1. Comment 1.

Sentence revised: lines 42-45

Page 2. Comment 1.  I suggest to use abbreviations after the first time.

Done

Comment 2.

I suggest the Authors to better clarify the meaning of methane emissions, climate change and livestock production.

Only enteric methane emissions were considered? or also the emissions coming from other sources at the farm level?

Which aspects of climate change were considered? Land use change, deforestation, water related issues or so on... Same for the meaning of livestock production.

I suggest the authors to make a table or chapter explaining those specifications.

The paragraph was revised: line 80-115.

Comment 3

The terms "climate change" include not only methane emissions but also other aspects correlated with livestock production such as land use change and so on. If in this review the hot topic is the methane emissions and strategy to reduce it, I suggest to focus it on "methane" instead of "climate change".

Dear reviewer. Thank you for your suggestion, which has been considered. The issue was considering how ruminant productivity is linked to climate change through methane and the links between ruminant production and the other aspects of climate change.

Page 4

Comment 1. Line

Done

Page 6

Comment 1

Both "livestock production" and "climate change" have a very wide meaning. I suggest the Authors to better specify the themes and topics that are included in the evaluation. Combining everything together seems a little vague.

Even in a supplementary table I suggest to summerize and report all the articles used and to cathegorize them into specific classes (e.g. "methane emissions"; "milk production and methane emissions"; "health and methane emissions")

It is also unclear if only entheric emissions are considered or also other GHGs emissions sources.

Page 7. the meaning of livestock production is vague

Page 9. the meaning of livestock production is vague

This comment was already considered. Lines 80-115.

Unclear sentence. Possible grammar/wrting mistake

the sentence has been revised

In the discussion the authors try to assess the different topics related to the meaning of climate change and methane emissions such as the application of different mitigation strategies or the interconnection between animal productivity and climate change. However, the result is slightly vague. I suggest that Authors to rewrite the discussion, or change some part of it, focusing on the evolution of the topics treated over time, as well as on the possible differences between countries. This will highlight the evolution of the topic as well as the differences in the needs of different countries.

The discussion was rewritten and merged to the results.

I suggest the Authors to insert a sentence to explain the complexity of those two topics, their meaning and all the aspects that might be included.

Done, please check lines 80 - 115.

I suggest the Authors to include some example or to show all the papers included with a sort of ranking.

Done

I suppose the Authors mean Green Feed system instead of green field

Yes. The sentence was revised.

why the Authors included other livestock? I think that the paper should be more focused on ruminants since the main target is methane. if not, a better explanation should be done in the materials and methods.

The reviewer is right. The sentence was revised. Explanation was also provided in the material and methods.

climate change is not only related to GHGs emissions.

Yes, we agree. Here, we first state one of the potential impacts of livestock production on the climate through methane. Then, the other aspects of climate change were considered (lines 402 - 4011).

conclusions are a bit long. Also some concepts already explained in the discussion are repeated. They should be summarized.

The conclusion was shortened

In the results those aspects are not included. I suggest the Authors to include a more specific chapter and discussion related to those points (e.g. which strategy received more attention, if there are differences over time or in different countries.). Also the relationship with livestock productivity have to be better explained. Which aspects did you consider? Which parameters of climate change are included?

These aspects have already been explained in the results and discussion section. We therefore remove the sentence from the conclusion.

Climate change was already explained (lines 80-115)

more references should be included. Maybe all the main reviews about climate change and livestock as well as all the paper included in the evaluation.

Dear Editor. We think it was unnecessary because the paper is not a narrative review but a review focusing on Bibliometrics within a given period.

Reviewer 3 Report

Comments and Suggestions for Authors

It is a helpful and useful review focus on the bibliometric mapping of academic research association with methane emissions and animal productivity.

There are only two suggestions.

1. You mention in the paper between 1987 and 2023 (in Line 24, 133, and etc), but from 1980 to 2023 (in Line 92, 94, 174, 194, and etc). please explain the difference. It bothered me and maybe also the readers.

2. It is better to describe the distribution about these 2500 documents. How many focus on cows? How many beef cattle? how many sheep, and how many goats? I think it is an interesting topic.

Author Response

Response to Reviewer 3

It is a helpful and useful review focus on the bibliometric mapping of academic research association with methane emissions and animal productivity.

Response. Thank you, dear reviewer, for your interest in our research.

There are only two suggestions.

  1. You mention in the paper between 1987 and 2023 (in Line 24, 133, and etc), but from 1980 to 2023 (in Line 92, 94, 174, 194, and etc). please explain the difference. It bothered me and maybe also the readers.

Response. The search focuses on two linkages and concentrates on 1980 – 2023. The results, however, show existing publications on “Methane emissions and animal productivity” from 1987 – 2023, and documents were funded on “Livestock production and climate change” from 1980 – 2023.

  1. It is better to describe the distribution about these 2500 documents. How many focus on cows? How many beef cattle? how many sheep, and how many goats? I think it is an interesting topic.

Response. Dear Editor. This is not the focus of this study; we want to see trends of existing publications on “Methane emissions and animal productivity” and “Livestock production and climate change,” not focusing on separating them based on ruminant species.